# Lung cancer symptoms awareness among Ethiopian adults: A latent class analysis

Nathan Estifanos[1,2]*, Gudina Egata[1], Adamu Addissie[1], Rahel Argaw Kebede[3], Amsalu Bekele[3], Negussie Deyessa[1]

1 School of Public Health, College of Health Sciences, Addis Ababa University, Addis Ababa, Ethiopia, 2 College of Medicine and Health Science, Wollo University, Dessie, Ethiopia, 3 College of Health Sciences, Addis Ababa University, Addis Ababa, Ethiopia

* estifanos9090@gmail.com

## Abstract

### Background

There is limited evidence regarding lung cancer awareness in developing countries. In Ethiopia, 92.2% of lung cancer patients present at facilities with late stages, leading to poor treatment outcomes. This emphasizes the importance of early detection. Symptom awareness is crucial for reducing delays. This study aimed to identify latent classes of lung cancer symptom awareness and their predictors, guiding class-specific interventions.

### Methods

A population-based cross-sectional survey was conducted from October to December 2023 among a randomly selected 2388 adults in Addis Ababa, Ethiopia. A face-to-face interview was conducted using the validated Lung Cancer Awareness Measure (Lung CAM). Latent class analysis and latent class multinomial logistic regression were used to identify classes and predictors of class membership.

### Results

Three distinct classes of participants were identified: "poor awareness" (Class 1: 38%), "fair awareness" (Class 2: 37.5%), and "good awareness" (Class 3: 24.5%). The average symptom awareness score was 7.8 out of 14. The most commonly recognized symptom was coughing up blood (72%), while changes in the shape of fingers were the least recognized (20%). Being male, employed, having a higher education level, using out-of-pocket money for health expenses, and knowing someone with cancer significantly increased the odds of belonging to the "good awareness" class, with adjusted odds ratios ranging from 1.66 to 12.60.

**Data availability statement:** All relevant data are within the paper and its Supporting information files.

**Funding:** Funder: Addis Ababa University Award Number: 6417 Grant Recipient: Nathan Estifanos Funder: Bristol Myers Squibb Foundation Award Number: 61329475 Grant Recipient: Nathan Estifanos The funders did not play a role in the design of the study, collection and analysis of data, decision to publish, or preparation of the manuscript.

**Competing interests:** The authors assert that they possess no conflicting interests.

## Conclusion

Only one-fourth of participants were classified as class 3, denoted as "good awareness," indicating a significant gap in symptom awareness. Respiratory symptoms were mostly well-known. Class membership varied across sociodemographic and related characteristics. Hence, there is a need for class-specific educational intervention and a focus on non-respiratory symptoms.

## Introduction

Lung cancer remains the leading cause of cancer morbidity and mortality globally in 2022 [1]. According to the Global Observatory of Cancer 2022 report, Eastern Africa has an age-standardized incidence rate of 3.9 and 2.7 per 100,000 for males and females, respectively [1]. It ranks among the top five most common cancers in men aged 15 and older, with an incidence rate of 3.5 per 100,000 in Ethiopia in 2015 [2].

In Ethiopia, tobacco use, an established risk factor, has been increasing [3]. Additionally, Ethiopia exceeds the World Health Organization (WHO) guideline [4] for the concentration of particulate matter < 2.5 microns [5], and around 95% of the population depends on polluting fuels [6]. The nation is one of the 30 countries facing a significant challenge with tuberculosis, alongside comorbidities like the human immunodeficiency virus and chronic pulmonary diseases [7,8].

Lung cancer symptoms include respiratory symptoms such as a persistent cough, chest pain, shortness of breath, and coughing up blood, as well as non-respiratory symptoms like unexplained weight loss, loss of appetite, shoulder pain, and changes in the shape of fingers/nails [9,10]. In Ethiopia, a significant proportion (92.2%) of lung cancer patients seek medical attention at health facilities during advanced stages [10]. The majority of patients (over 90%) diagnosed with lung cancer exhibit symptoms at the time of diagnosis [11]. Notably, cough is the most prevalent symptom [12].

Early diagnosis is widely agreed to improve treatment outcomes [13,14]. In countries lacking organized population-based screening programs, like Ethiopia, it is crucial to raise awareness of symptoms to reduce delays in both diagnosis and treatment [15,16]. Accordingly, Ethiopia's cancer control plan [17] has prioritized increasing symptom awareness as a key strategy to achieve Sustainable Development Goal (SDG) 3.4 [18], which focuses on reducing premature mortality from non-communicable diseases through prevention and treatment. However, most studies assessing population-level awareness of lung cancer symptoms have been conducted in high-income countries [12,19–21], with limited evidence available from low- and middle-income countries (LMICs) [16,22–28]. Moreover, existing studies used arbitrary cutoffs to define "good" awareness, including the widely used Lung CAM, which lacks standard thresholds [12,19,20,22,24–28]. In contrast, latent class analysis (LCA), a data-driven method that identifies subgroups based on symptom awareness and offers a better alternative to conventional classification approaches for guiding targeted interventions [29–34].

To the best of the researchers' knowledge, no prior study was done on lung cancer symptom awareness in Ethiopia. Hence, this study aimed to identify latent classes of lung cancer symptom awareness and their predictors to guide class-specific interventions.

## Methods

### Study setting, period, and design

A population-based cross-sectional survey was done in Addis Ababa, Ethiopia, from October 30 to December 29, 2023. It was a part of a multinational lung cancer diagnosis and control project in Ethiopia from 2020 to 2023. The project was implemented by seven implementers in collaboration with the Ministry of Health of Ethiopia. Mathiwos Wondu-YeEthiopia Cancer Society was the lead implementer, and the Ethiopian Thoracic Society (ETS) executed the research part of the project. Addis Ababa is Ethiopia's capital and largest city, home to an estimated 4.5 million people, 68% of whom are adults [35]. It comprises eleven sub-cities, which are divided into over 100 districts [36]. Addis Ababa is also a referral site, housing the first and largest chest unit and radiotherapy center located at Tikur Anbessa Specialized Hospital.

### Study participants

The source population for this study comprised all adults aged 18 years and above residing in Addis Ababa. The study population consisted of a randomly selected sample of adults who had lived in the city for at least six months. Individuals with mental illness or severe illness that prevented them from participating in the interview were excluded from the study.

### Sample size determination and sampling procedure

A total of 2341 eligible adults from selected households participated in the study. The determination of the sample size was based on Cochran's single population proportion formula [37], with a 95% confidence interval and a 3% margin of error. Since the proportion of lung cancer awareness in our context was unknown, a conservative assumption of P = 50% was made. Additionally, a design effect of 2 and a non-response rate of 15% were considered, resulting in a final sample size of 2388 participants. A large sample size is required for the LCA approach, as models and fit statistics have shown high accuracy when the sample size exceeds 500 on average [30,31].

Participants for the study were enrolled using a multistage stratified cluster sampling method. A list of all census enumeration areas (EAs) in Addis Ababa, prepared for the 2019 population and housing census by the central statistical agency, was used as a sampling frame [38]. From this list, 96 enumerator areas were randomly selected from the eleven sub-cities of Addis Ababa, considering their sizes. Then, 25 households were selected from each EAs using systematic random sampling. Within each chosen household, one eligible adult was interviewed. If there were multiple eligible adults in a selected household, the lottery method was used to choose one. Any adults who declined to participate or could not be reached after multiple attempts were not replaced by adults from neighbouring households.

### Data collection tools and measurement

**Measurement of the dependent variable.** The tools we have used have two sections. The first section of the questionnaire was used to measure lung cancer symptoms awareness, which is a latent variable in our study. In LCA, since observable indicators are the main factors influencing class features, a compelling argument should be presented for their inclusion in the measurement models. Therefore, the research question should determine the most of the indicators used for the analysis, and the selection of indicator variables should be guided by theory [29,31]. The optimal number of indicator variables to incorporate in a model remains a topic of discussion; however, increasing their number generally leads to improved results. Some studies have utilized up to twenty indicators, while others have relied on only four indicators [31,39].

In this study, we assessed the latent variable of lung cancer symptom awareness by using a set of 14 indicator variables adopted from the validated Lung CAM [19]. The 14 indicator variables were actual symptoms of lung cancer, presented in the form of close-ended questions with yes/no/don't know options. Participants were asked to check whether they recognized them or not. The questionnaire was translated and adapted by the World Health Organization's recommendations [40]. The original Lung CAM was translated into Amharic by two bilingual healthcare experts. Following that, two additional bilingual healthcare experts conducted a back-translation of the Amharic version into English. To ensure the content validity and accuracy of the translation, 10 experts, including 5 pulmonologists and 5 oncologists, reviewed the Amharic version of the tool. Based on their review and rating, the overall content validity index was 0.96, which is acceptable.

Before commencing data collection, the translated version of the questionnaires was piloted on 100 individuals to assess its clarity and reliability. The overall reliability of the 14 indicators was checked using Cronbach's alpha, yielding a coefficient of 0.94.

**Measurement of independent variables.** The second section of the questionnaire was a validated structured questionnaire. It addressed the sociodemographic, behavioral, clinical, and related characteristics of the study participants. A subset of Ethiopian demographic and health survey tools is used to collect information on the socio-demographic and household economic status of participants. Socio-demographic characteristics such as gender, age, marital status, religion, educational status, occupational status, and wealth index were included. Principal Component Analysis, as indicated in the 2019 Ethiopian Demographic and Health Survey, was used to classify participants based on their wealth index into five quantiles [38].

The study assessed the participants' behavioral characteristics related to cigarette smoking, exposure to second-hand smoke, shisha smoking, and alcohol drinking history. The assessment involved categorizing participants into three groups: current, ex (former), and never. Finally, for analysis purposes, the responses were further classified into two categories: "ever" and "never," by combining current and former responses into the "ever" category [19,20,22,23]. This was done due to the low prevalence of smoking—particularly among females—to avoid small cell sizes and ensure more stable estimates.

Clinical and related characteristics: In the study, clinical and related characteristics were assessed using the following questions and criteria.

To assess the presence of chronic diseases, participants were asked, "Have you ever been told by a healthcare professional that you have any of the listed medical conditions?" They were presented with a list of chronic diseases, including Hypertension, Diabetes Mellitus, Asthma, Bronchitis, and Cancer, along with other options. If a participant answered "Yes" to any of the listed chronic diseases, they were classified as having a chronic disease. On the contrary, if a participant answered "No" to all the diseases listed, they were classified as not having chronic diseases [28].

Two variables related to familiarity with cancer were assessed. Participants were asked, "Do you know someone with lung cancer?" and "Do you know someone with other types of cancer?" They were given the option to respond with a "Yes" or "No" for each question [19,22,27,28].

The payment option for medical care was determined by asking participants, "How do you usually pay for medical care?" They were provided with options such as out-of-pocket money, community-based health insurance, family and partner support, and an option to specify "Other" [22]. The survey questionnaires are attached as a supporting document (S1 File).

Data collection was conducted through face-to-face interviews using the KoboToolbox software. The interviews were conducted by 18 trained nurses and health officers, each holding a bachelor's degree and served as data collectors, supervised by 9 individuals with master's degrees in public health.

## Data quality assurance

Various strategies were implemented to ensure the validity and reliability of the survey. The survey questionnaires underwent an adoption and validation process. Extensive training sessions were organized to equip data collectors and

supervisors with the research objectives and standardize their interviewing approach. Moreover, both data collectors and supervisors engaged in role-plays and pre-test exercises before the actual data collection commenced. A standardized operational manual was exclusively developed for data collectors to provide explicit directives and instructions. The entire data collection process was closely supervised. Both the principal investigators and supervisors are involved in the supervision process. Consequently, any identified errors were promptly addressed in the field. Furthermore, quality assurance measures were implemented during the survey's data management phase to ensure the collected data's accuracy and reliability. Data validation and cleaning were performed to identify and resolve any inconsistencies, errors, or missing values in the dataset, as well as to address any anomalies or outliers.

## Data processing and analysis

The data collection was conducted using the KoboToolbox, followed by exporting it as an Excel file and performing data cleaning using SPSS version 26. Stata version 16.1 was used for the analysis. There were no missing data for the variables included in this analysis. To describe the socio-demographic, behavioral, clinical, and related characteristics of participants descriptive statistics were done. To identify the most and least recognized symptoms of lung cancer, percentages were calculated for the 14 indicators of lung cancer symptoms.

The characteristics of the indicator variables and the latent variable are closely interconnected. To generate a set of mutually exclusive latent classes, which are groups of individuals, LCA is used to analyze patterns of responses to the indicator variables [31]. In our study, participants were classified based on their awareness of the 14 symptoms of lung cancer. To achieve this, the 14 lung cancer symptoms awareness indicator variables were coded as binary variables, either as "yes" or "no (including don't know)," and included in the latent class measurement model. We started fitting a one-class model and continued fitting subsequent models until we identified the best-fit model, as recommended [29,31,41]. To obtain the global maximum log-likelihoods for each model, we reran the model 200 times with multiple random starts. Since there is no single indicator that can accurately determine the best model fit, the selection of models was made by considering a combination of simplicity, ease of interpretation, and various fit indices [29,31,41].

The fitness of the models was assessed using various metrics, including the Akaike Information Criterion (AIC) and the Bayesian Information Criterion (BIC). The better the model, the lower the AIC and BIC values [31]. Comparing the fitted model with a saturated model is one method used to evaluate the model fit in standard latent class models, where all observable variables are categorical. The likelihood-ratio test ($G^2$) was utilized in this investigation to ascertain the degree to which our model matched the saturated model. It is a single-model goodness-of-fit measure. An insignificant likelihood-ratio test ($G^2$) p-value (> 0.05) indicates that our model fits the data as well as the saturated model. This suggests that our model effectively represents the observed data [42].

The evaluation of classification diagnostics involved the utilization of entropy and the average latent class posterior probability. A higher entropy value closer to 1 indicates better separation of latent classes, while average latent class posterior probability values closer to 1.0 suggest an accurate prediction of class membership. Acceptable values for entropy and average latent class posterior probability are those above 0.8. However, there are currently no established criteria for determining the optimal size of latent classes. Latent classes comprising less than 5 percent of the sample are less desirable. When considering the complexity of the model, a simpler model is preferred [31].

Statistical criteria must be assessed alongside interpretability to ensure meaningful results. A class solution may have excellent statistical properties, but it is not beneficial if it lacks theoretical relevance [43]. Therefore, the identification of the optimal class solution was based on both the theoretical and practical implications of the latent classes. We conducted a conditional analysis using Pearson correlation coefficients to examine the violation of the local independence assumption among the 14 indicators within each class. Finally, we carefully evaluated and appropriately presented the crucial parameter estimates for fitting the optimal LCA model.

The latent class memberships were anticipated through the latent class multinomial logistic regression model after the enumeration of latent classes. Using assigned latent class membership as dependent variables in multinomial logistic regression is prone to amplifying the impact of predictors which is known as the two-step approach.

After latent class enumeration, the class memberships were predicted using a latent class multinomial logistic regression model. Using assigned latent class membership as a dependent variable in multinomial logistic regression is prone to amplifying the impact of predictors, which is known as the two-step approach [44]. Therefore, after fixing the class solution, a one-step technique was employed, in which predictor variables were added to the LCA model [45]. The unadjusted latent class multinomial logistic regression model underwent fitting with each independent variable individually to determine their association with latent class membership. The crude odds ratio along with a 95% confidence interval (CI) and corresponding p-value were calculated. The final adjusted latent class multinomial logistic regression model was fitted with the independent variables that showed a significant association (P-value < 0.25) in the unadjusted model. In the final model, significant associations were declared with a p-value of less than 0.05. Multi-collinearity was checked by using variance inflation factor and tolerance tests.

The result was finally standardized by accounting for both the design weight and post-stratification weight. To account for the complex survey design, we estimated the design weight based on the inverse of the selection probability of study participants. To ensure that our study participants closely resemble the source population, we calculated post-stratification weights. Poststratification weight is estimated to depend on the sex and age structure of the population in Addis Ababa. The final weight was calculated by multiplying the two weights [35]. Finally, the results were presented using tables, graphs, and figures. The reporting was guided by a recommendation of the STROBE checklist for reporting observational studies (S1 Checklist).

### Ethics statement

Ethical approval was granted by the institutional review board of Addis Ababa University, College of Health Sciences, with protocol number 073/23/SPH. Additionally, a formal supporting letter was obtained from the Addis Ababa Health Office. Approval was successfully obtained from all administrative levels within Addis Ababa. Following a comprehensive description of the study's objectives, potential benefits, and confidentiality measures, written consent was obtained from all participants.

## Results

### Socio-demographic characteristics of the study participants

Out of the 2,388 adults invited to participate in the study, 2,341 responded, resulting in a response rate of 98%. The 47 non-responders were unavailable at home despite three repeated visits. The study population was predominantly young and educated, with just over half being female. Wealth levels varied, with about one-fifth of participants classified in the poorest category (Table 1).

### Behavioral and clinical characteristics of the study participants

About 6% of participants had a history of cigarette smoking, and most (78%) relied on out-of-pocket payments for healthcare. Familiarity with cancer was limited—fewer than 2% knew someone with lung cancer, and 11% knew someone with other cancers (Table 2).

### Recognition of lung cancer symptoms

To assess symptom recognition, participants were asked about 14 lung cancer symptoms. Recognition levels varied, with symptoms such as coughing up blood being recognized by most participants, while changes in the shape of fingers or nails were the least recognized. On average, participants identified 7.8 symptoms (SE = 0.2; 95% CI: 7.4–8.2) (Fig 1).

**Table 1. Socio-demographic characteristics of survey participants, Addis Ababa, Ethiopia,2023.**

| Variables | Categories | Frequency(n = 2341) | Percent (%) |
|---|---|---|---|
| **Sex** | Male | 1129 | 48.3 |
| | Female | 1212 | 51.7 |
| **Age** | 18-29 | 1191 | 50.9 |
| | 30-39 | 540 | 23.0 |
| | 40-49 | 286 | 12.2 |
| | 50-59 | 170 | 7.3 |
| | ≥ 60 | 154 | 6.6 |
| **Marital status** | Single | 936 | 39.98 |
| | Married | 1229 | 52.5 |
| | Widowed | 105 | 4.49 |
| | Divorced | 58 | 2.49 |
| | Others | 13 | 0.54 |
| **Religion** | Christian | 1940 | 82.9 |
| | Muslim | 345 | 14.8 |
| | Others | 56 | 2.3 |
| **Level of education** | Unable to read and write | 84 | 3.6 |
| | Able to read and write | 168 | 7.2 |
| | Primary | 468 | 20.0 |
| | Secondary | 715 | 30.5 |
| | Diploma and vocational | 367 | 15.7 |
| | Degree and above | 539 | 23.0 |
| **Occupation** | Employed | 543 | 23.1 |
| | Merchant | 354 | 15.1 |
| | Student | 974 | 41.6 |
| | Unemployed | 330 | 14.1 |
| | Housewife | 91 | 4.0 |
| | Others | 49 | 2.1 |
| **Wealth index** | Poorest | 488 | 21.0 |
| | Poorer | 475 | 20.0 |
| | Rich | 444 | 19.0 |
| | Richer | 418 | 18.0 |
| | Richest | 516 | 22.0 |

**Note:** Others-Retirees and Farmers.

### Fitting the LCA model

Accurately determining the number of latent classes and correctly assigning individuals to their respective classes with a high level of confidence is a crucial task in LCA [29,31]. The LCA measurement model was fitted using fourteen indicators of lung cancer symptom awareness. Because the number of latent classes is unknown a priori, iterative processes are used in LCA to determine the appropriate number of classes. Accordingly, models with one to six latent classes were fitted. Both theoretical and statistical criteria were applied to find the model that best fits the data. Model fit was evaluated using AIC, BIC, likelihood-ratio tests, entropy, and class size. The three-class model showed lower AIC and BIC values than the one- and two-class models, indicating a better fit. The likelihood-ratio test (G²) p-value exceeded 0.05, suggesting an adequate fit to the data. Additionally, the three-class model demonstrated higher entropy, indicating better class

**Table 2. Behavioral and Clinical Characteristics of survey participants, Addis Ababa, Ethiopia, 2023.**

| Variables | Category | Frequency(n=2341) | Percent (%) |
|---|---|---|---|
| **Ever smoked cigarettes** | Yes | 148 | 6.3 |
| | No | 2193 | 93.7 |
| **Ever smoked shisha** | Yes | 91 | 3.9 |
| | No | 2250 | 96.1 |
| **Ever drank alcohol** | Yes | 857 | 36.6 |
| | No | 1484 | 63.4 |
| **Having a chronic disease** | Yes | 370 | 15.8 |
| | No | 1971 | 84.2 |
| **Payment option for medical service** | CBHI users | 455 | 19.4 |
| | Family and partner support | 32 | 1.4 |
| | Out pocket money | 1827 | 78.0 |
| | Others | 27 | 1.2 |
| **Knowing someone with lung cancer** | Yes | 40 | 1.7 |
| | No | 2301 | 98.3 |
| **Knowing someone with other types of cancer** | Yes | 249 | 10.7 |
| | No | 2092 | 89.3 |

**Note:** CBHI-community-based health insurance, Others- Private and employer sponsor health insurance.

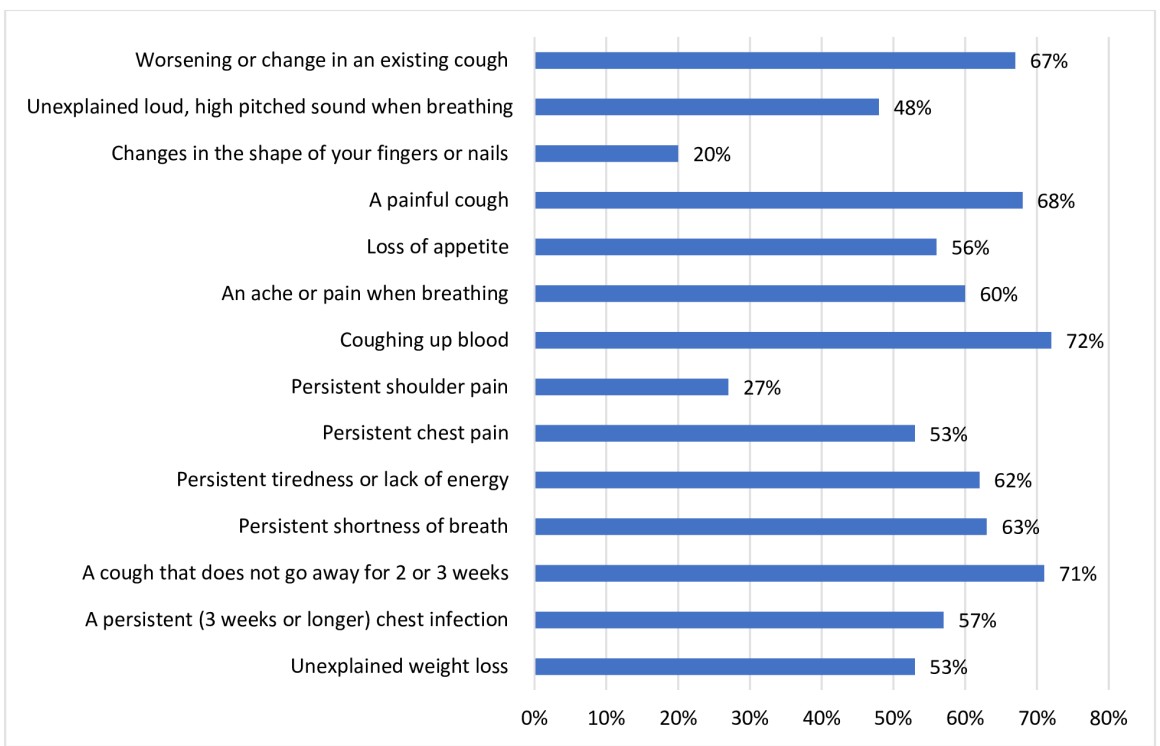

**Fig 1. Awareness of lung cancer symptoms among Ethiopian Adults, Addis Ababa, Ethiopia, 2023.**

**Table 3. Each class fitted in the LCA model with model fit statistics, Addis Ababa, Ethiopia, 2023.**

| Model comparison | | | | | | | | Latent class size based on model assignment | | | | | |
|---|---|---|---|---|---|---|---|---|---|---|---|---|---|
| Class | AIC | BIC | Log-likelihood | df | G² | G² P | Entropy | LC1 | LC2 | LC3 | LC4 | LC5 | LC6 |
| 1 | 42435 | 42516 | −21204 | 14 | 17750 | 0.00 | 1 | 100 | | | | | |
| 2 | 29401 | 29568 | −14671 | 29 | 4686 | 1.00 | 0.985 | 39 | 61 | | | | |
| 3 | **28701** | **28955** | **−14307** | **44** | **3956** | **1.00** | **0.860** | **37.9** | **37.5** | **24.6** | | | |
| 4 | 28173 | 28513 | −14028 | 56 | 3398 | 1.00 | 0.784 | 11 | 27 | 38 | 24 | | |
| 5 | 28064 | 28490 | −13958 | 70 | 3259 | 1.00 | 0.779 | 11 | 27 | 16 | 21 | 25 | |
| 6 | 28000 | 28512 | −13911 | 82 | 3164 | 1.00 | 0.785 | 11 | 27 | 6 | 18 | 12 | 26 |

**Note:** AIC = Akaike Information Criterion, BIC = Bayesian Information Criterion, G² = Likelihood ratio statistic, G²P = model vs. saturated p > chi2, LC = Latent class.

separation, and acceptable average posterior probabilities (all above 0.8), with the smallest class comprising more than 5% of the sample. Although adding more classes improved model fit based on AIC and BIC, it compromised class separation and increased model complexity, making the three-class solution preferable. Importantly, the three-class solution yielded theoretically meaningful classes, as the item response probabilities across the classes allowed for straightforward labeling. Therefore, three classes were determined to be the optimal solution for our data (Table 3).

## Latent class size and item response probabilities across the three classes

The latent class membership probability was computed to determine the size of the latent classes. Based on the three-class solution, 37.9% of the participants belong to class 1, while class 2 comprises 37.5% of the participants. The remaining 24.6% of participants are classified under Class 3. The item response probabilities display the variation in response patterns across the three classes that guide differentiation between classes. We identified the general meaning of the three classes by examining the item response pattern across them, allowing us to assign appropriate labels to each one. Both the weighted and unweighted item response probabilities for the three classes are attached as supporting documents (S1 Table). Overall, our item response probability showed that participants in Class 3 demonstrated a considerably greater probability of recognizing the 14 lung cancer symptoms, whereas those in Class 1 exhibited a lower likelihood of identifying these symptoms. The 14 symptoms of lung cancer were fairly recognized by the participants in class 2. Therefore, Class 1, Class 2, and Class 3 were designated as the "low awareness class," "fair awareness class," and "good awareness class," respectively (Fig 2).

## Predictors of latent class membership

Latent class multinomial logistic regression was done to identify predictors of latent class membership. In the final adjusted latent class multinomial logistic regression model, only sex, age, level of education, occupation, wealth index, payment option for medical care, ever smoked cigarettes, having a chronic disease, knowing someone with lung cancer, and knowing someone with other types of cancers were retained. Class 1, the "Low awareness class," was the reference category in both adjusted and unadjusted models. A significance threshold of P < 0.05 was employed to establish statistical significance in the adjusted model. The results of the weighted final latent-class multinomial logistic regression analysis are presented in Table 4, while the unweighted final latent-class multinomial logistic regression results are provided as a supporting file (S2 Table).

Based on the final weighted latent-class multinomial logistic regression, compared to the Class 1: "low awareness" class, members of the Class 3: "good awareness" class were more likely to be male [(AOR = 1.66, 95% CI:(1.13,2.43)] and employed [(AOR = 1.85, 95% CI: (1.13,3.04)].

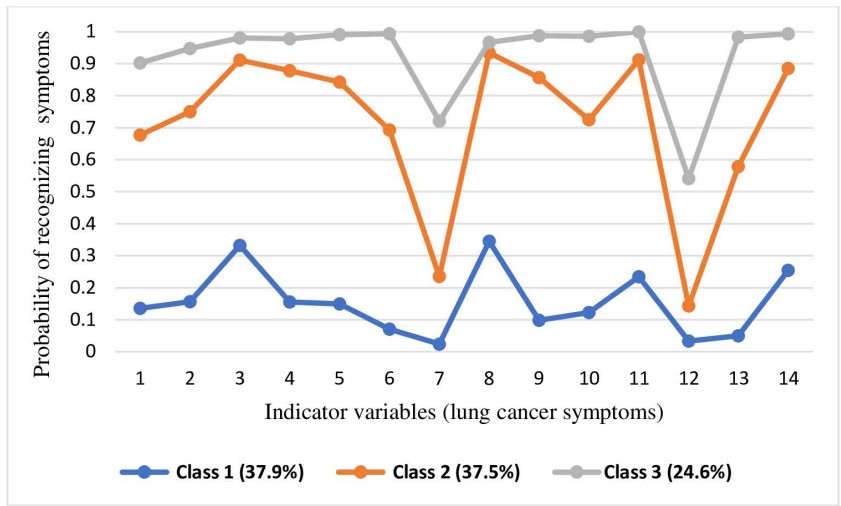

**Fig 2. The class-specific probabilities of correctly recognizing lung cancer symptoms, for the three latent classes of study participants, Addis Ababa, Ethiopia, 2023. Indicator variables (lung cancer symptoms) 1.** Unexplained weight loss **2.** A persistent (3 weeks or longer) chest infection **3.** A cough that does not go away for two or three weeks **4.** Persistent shortness of breath **5.** Persistent tiredness or lack of energy **6.** Persistent chest pain **7.** Persistent shoulder pain **8.** Coughing up blood **9.** An ache or pain when breathing **10.** loss of appetite **11.** A painful cough **12.** Changes in the shape of your fingers or nails (Finger clubbing) **13.** Developing an unexplained loud, high-pitched sound when breathing (Stridor) **14.** Worsening or change in an existing cough.

Participants with higher education were more likely to be categorized in the "good or fair awareness" class than in Class 1: "low awareness" class, compared to those who were unable to read and write. Concerning the wealth index, the participants who were classified as richer and richest had odds that were 1.8 [(AOR = 1.80, 95% CI:(1.03,3.12)] and 1.64 [(AOR = 1.64, 95% CI:(1.02,2.63)] times higher, respectively, than those who were classified as the poorest group.

Compared to class 1: the "low awareness" class, members of the "fair and good awareness" class were significantly more likely to cover their medical expenses out of pocket. For Class 2, the AOR for "good awareness" was 1.54 [(AOR = 1.54, 95% CI: (1.10, 2.20)], and for Class 3, the AOR for "good awareness" was 2.12 [(AOR = 2.12, 95% CI: (1.30, 3.46)].

The odds of being placed in class 3: "good awareness" were 2.94 [(AOR = 2.94, 95% CI: (1.16, 7.45)] times higher for participants who knew someone with lung cancer than for those in the reference category (do not know someone with lung cancer) when compared to those in class 1: "low awareness". Compared to the "low awareness" class, members of the "good and fair awareness" classes were significantly more likely to know someone with other types of cancer. For Class 2: "fair awareness," the AOR was 1.65 [(AOR = 1.65, 95% CI: (1.02, 2.68)], and for Class 3: "good awareness," the AOR was 2.54 [(AOR = 2.54, 95% CI: 1.69, 3.83)].

Based on our study results, age, ever-smoked cigarettes, and having a chronic disease did not predict class membership (Table 4).

## Discussion

Even though survival and effective treatment depend on early detection, over 90% of Ethiopian lung cancer patients arrived at facilities in advanced stages [10]. Being aware of the symptoms is essential to minimizing delays [11]. However, evidence on lung cancer awareness in LMICs is limited. Thus, basic evidence that aids in the design of a customized group intervention to reduce late presentation is needed. Using latent class analysis of 2,341 adults in Addis Ababa, we identified three distinct awareness groups. Only one-fourth had good awareness, with respiratory symptoms more

**Table 4. Latent class multinomial logistic regression of predictors of latent class membership, Addis Ababa, Ethiopia,2023.**

| Variables [1] | Class 2: "Fair awareness class" [1] | | Class 3: "Good awareness class" [1] | |
|---|---|---|---|---|
| | AOR (95% CI) | P-Value | AOR (95% CI) | P-Value |
| **Sex** | | | | |
| Male | 1.18 (0.83, 1.67) | 0.36 | 1.66(1.13,2.43) | **0.001** |
| Female | 1 | | | |
| **Age (years)** | | | | |
| 30-39 | 0.97 (0.71, 1.32) | 0.86 | 1.03(0.68,1.57) | 0.89 |
| 40-49 | 0.88 (0.60,1.30) | 0.52 | 1.03(0.64,1.65) | 0.91 |
| 50-59 | 1.5 (0.90, 2.38) | 0.12 | 1.42(0.80,2.55) | 0.24 |
| ≥60 | 0.94 (0.53,1.64) | 0.81 | 1.00(0.44,2.28) | 0.99 |
| 18-29 | 1 | | 1 | |
| **Level of education** | | | | |
| Able to read and write | 0.74(0.41,1.32) | 0.31 | 0.64(0.17,2.35) | 0.50 |
| Primary education | 0.80 (0.45,1.44) | 0.46 | 3.07(0.90,10.6) | 0.08 |
| Secondary education | 1.84 (1.06,3.20) | **0.03** | 5.70(1.67,19.5) | **0.006** |
| Diploma and vocational | 1.94(1.00,3.77) | **0.05** | 7.46(2.14-25.9) | **0.002** |
| Degree and above | 2.74 (1.43,5.26) | **0.003** | 12.6 (3.88,40.8) | **<0.001** |
| Unable to read and write | 1 | | 1 | |
| **Occupation** | | | | |
| Employee | 1.24 (0.76,2.03) | 0.39 | 1.85(1.13,3.04) | **0.015** |
| Merchant | 0.93 (0.61,1.41) | 0.73 | 0.71(0.42,1.21) | 0.21 |
| Student | 1.00 (0.54,1.83) | 0.99 | 1.27(0.44,3.61) | 0.65 |
| Unemployed | 1.53 (0.71,3.32) | 0.28 | 1.31(0.53,3.26) | 0.55 |
| Others[1] | 1.14(0.60,2.17) | 0.68 | 0.39(0.11,1.36) | 0.14 |
| Housewife | 1 | | 1 | |
| **Wealth index** | | | | |
| Poorer | 1.20(0.80,1.80) | 0.36 | 1.02(0.60,1.74) | 0.94 |
| Medium | 1.29(0.74,2.25) | 0.37 | 1.70(0.94,3.05) | 0.08 |
| Richer | 1.80(1.03,3.12) | **0.04** | 1.76(1.00,3.14) | 0.05 |
| Richest | 1.64(1.02,2.63) | **0.04** | 1.79(0.93,3.44) | 0.08 |
| Poorest | 1 | | | |
| **Payment option for medical services** | | | | |
| Family and partner support | 0.39(0.13,1.18) | 0.10 | 0.64(0.14,2.84) | 0.55 |
| Out pocket money | 1.54(1.10,2.20) | **0.01** | 2.12(1.30,3.46) | **0.003** |
| Others[2] | 3.10(0.65,14.7) | 0.15 | 2.82(0.38,21.0) | 0.31 |
| CBHI users | 1 | | | |
| **Ever smoked cigarettes** | | | | |
| Yes | 0.82(0.38,1.73) | 0.59 | 1.14(0.59,2.17) | 0.70 |
| No | 1 | | | |
| **Having a chronic disease** | | | | |
| Yes | 0.86(0.62,1.20) | 0.38 | 0.95(0.54,1.70) | 0.85 |
| No | 1 | | | |
| **Knowing someone with lung cancer** | | | | |
| Yes | 1.15(0.40,3.33) | 0.80 | 2.94(1.16,7.45) | **0.02** |
| No | 1 | | | |

*(Continued)*

**Table 4.** (Continued)

| Variables [1] | Class 2: "Fair awareness class" [1] | | Class 3: "Good awareness class" [1] | |
| --- | --- | --- | --- | --- |
| | AOR (95% CI) | P-Value | AOR (95% CI) | P-Value |
| **Knowing someone with other types of cancer** | | | | |
| Yes | 1.65(1.02,2.68) | **0.04** | 2.54(1.69,3.83) | **<0.001** |
| No | 1 | | 1 | |

**Note:** [1]: Class 1: "Low awareness" class was considered as a reference category model, Others[1]: Retirees and farmers, Others[2]: Private and employer sponsor health insurance, CBHI: Community-based health insurance.

recognized than non-respiratory ones. Predictors of good awareness included being male, educated, employed, using out-of-pocket money for healthcare, and knowing someone with cancer.

The person-centered LCA approach enabled the identification of three distinct awareness subgroups, highlighting the heterogeneity in public awareness of lung cancer symptoms. Similar three-class solutions were reported in breast cancer awareness studies from Iran [32,33], suggesting that LCA can effectively capture awareness heterogeneity in LMICs settings. This methodological advancement allows for the design of tailored, class-specific interventions [30,31], addressing limitations of previous studies that relied on arbitrary cutoff points for classification [20,22,23,26–28,46,47]. The relatively low proportion of participants in the "good awareness" group, compared to studies of awareness levels reported for other cancers in Ethiopia—67.3% for general cancer, 64.3% for breast, and 42.4% for colorectal [47–49]—signals a substantial gap in public awareness of lung cancer. This is partly explained by the little attention given by policymakers [50]. However, direct comparisons between studies should be interpreted with caution due to methodological differences, particularly in the grouping techniques used (LCA vs. arbitrary cutoff points).

As expected, our findings revealed symptom-specific variations: respiratory symptoms were more commonly recognized than non-respiratory ones, consistent with prior studies [19,20,22,25,26,28]. Hemoptysis emerged as the most recognized symptom, aligning with findings from the UK [19], Ireland [21], Nigeria [26], and South Africa [16]. However, hemoptysis occurs in only about 20% of lung cancer patients [51], which is also the case in the Ethiopian context [10]. Moreover, it is twice as common in smokers than non-smokers [9], yet only 25% of lung cancer patients in Ethiopia are smokers [10]. Therefore, relying on awareness of this symptom alone is insufficient for early detection in the Ethiopian context. On the other hand, symptoms such as changes in the shape of fingers or nails—although strongly predictive of lung cancer [11], were among the least recognized. This trend has been observed across several countries [19,21,26]. These findings highlight the importance of increasing public awareness about lesser-known but clinically significant symptoms of lung cancer.

Several sociodemographic factors were found to predict class membership. The relationship between awareness of lung cancer symptoms and sex remains inconclusive, as previous studies have reported mixed patterns [12,19,20,22,26,28,46,47]. In our study, however, men were more likely to belong to the good awareness class. These inconsistent findings may reflect underlying sociocultural differences. In the Ethiopian context, men often spend more time outside the home engaging in activities such as learning, socializing, and traveling, which may increase their exposure to health information [52]. Further research is needed to establish a clear relationship that can inform the development of tailored awareness interventions.

Unsurprisingly, higher educational status predicted better awareness, consistent with findings from several studies [26–28,46,47]. This well-established association likely reflects the role of education in enhancing access to health information, improving health literacy, and creating opportunities for health-related discussions through formal settings and digital platforms [53,54]. Similarly, employment was associated with higher lung cancer symptom awareness, a finding that aligns with studies from Saudi Arabia [20] and Palestine [28]. This may be attributed to increased exposure to health information,

workplace safety education, and better access to healthcare services. In urban and industrial settings like Addis Ababa [55], certain occupations—such as those in construction or manufacturing—may further contribute to heightened awareness due to occupational exposure and training related to cancer risks [22,56]. These findings underscore the importance of targeting uneducated and unemployed groups in awareness-raising interventions.

The association between socioeconomic status and lung cancer symptom awareness observed in our study aligns with previous reports [19,26–28,46]. It is well established that individuals with higher socioeconomic status tend to have easier access to health information through various media platforms and better healthcare services, including private providers—especially in urban settings like Addis Ababa. This increased access likely facilitates better awareness and earlier recognition of symptoms. Similarly, participants who paid out-of-pocket for healthcare demonstrated better awareness than CBHI users. This may reflect systemic gaps in the quality and accessibility of health service within the CBHI framework. Reports of limited medication availability, long waiting times, and poor service quality among CBHI users [57,58], may hinder their exposure to essential health information. These disparities underscore the need for targeted educational efforts within underserved communities, particularly those relying on public or subsidized health systems.

Familiarity with cancer, gained through knowing someone with lung cancer or other types of cancer, predicted membership in the good awareness class, a finding supported by studies in the UK [19], Palestine [28], India [27], and Nigeria [26]. Such familiarity may increase attention to symptoms and encourage information-seeking behaviors [59], particularly in caregiving contexts. However, this effect was limited in our setting [50], where only 0.17% of participants had such direct exposure, underscoring the broader need for community-based education. Additionally, educational campaigns could benefit from incorporating patient stories or real-life experiences to raise awareness. One unexpected finding was the lack of a significant difference in awareness between smokers and non-smokers. This may be attributed to the low smoking prevalence in the general population. One unexpected finding was the lack of significant difference in awareness between smokers and non-smokers. This may be attributed to low smoking prevalence in the general population [3] or insufficient targeting of smokers in awareness campaigns. It also suggests that both groups have limited awareness, reinforcing the need for widespread public education.

To the best of the researchers' knowledge, this study is the first population-based survey in Ethiopia assessing awareness of lung cancer symptoms, making it a significant contribution to the field. Using LCA, the study identified distinct awareness profiles that can guide targeted educational interventions. In addition, the large sample size, use of the pretested validated tool, face-to-face interview, standard enumerator areas, consideration of complex survey design, and standardized results using survey weight are the strengths of this study. However, the study has some limitations. Due to its cross-sectional design, causal relationships cannot be established. Furthermore, the study was limited to Addis Ababa, so the findings may not be generalizable to rural areas. Nevertheless, given the city's relatively better access to health information and services, the low awareness observed here likely indicates even lower awareness in less-served regions. Additionally, as in many studies [12,19,20,22,24–28], the outcome was measured using validated closed-ended (prompted) questions from the Lung CAM which may lead to overestimation by giving a chance of guessing. However, recognition of symptoms through prompted questions is often more relevant to health-seeking behavior [60], as individuals may recognize symptoms when experienced even if they do not spontaneously recall them.

The findings contribute to policy and practice by providing a baseline for targeted awareness strategies. The results support Ethiopia's National Cancer Control Plan, which prioritizes increasing cancer awareness as part of early detection efforts [17], and support achievement of SDG 3.4 [18] on reducing premature mortality from NCDs. Educational materials tailored to latent class profiles could improve recognition of less-known symptoms and reduce diagnostic delays. As lung cancer awareness includes symptoms, risk factors, and screening, future studies should explore these dimensions and extend research beyond urban settings.

## Conclusion

Only one-fourth of participants demonstrated good awareness of lung cancer symptoms, highlighting a significant gap in awareness, especially of non-respiratory symptoms. Awareness varied by sociodemographic factors such as sex, education, employment, wealth, health spending, and familiarity with cancer. These findings underscore the urgent need for tailored, class-specific educational interventions that target less-recognized symptoms to promote earlier detection and improve lung cancer outcomes in Ethiopia.

## Supporting information

**S1 File.  Lung cancer symptoms awareness survey tool.**
(DOCX)

**S1 Checklist.  STROBE Statement—checklist of items that should be included in reports of observational studies.**
(DOCX)

**S1 Table.  Weighted and Unweighted latent class marginal means of lung cancer symptoms awareness.**
(DOCX)

**S2 Table.  Unweighted latent class multinomial logistic regression of predictors of class membership.**
(DOCX)

## Acknowledgments

We would like to thank the Ethiopian Thoracic Society for their participation in the administration and execution of this project. The study participants and data collectors who enthusiastically engaged in this research are greatly appreciated by the authors.

## Author contributions

**Conceptualization:** Nathan Estifanos.

**Data curation:** Nathan Estifanos.

**Formal analysis:** Nathan Estifanos.

**Funding acquisition:** Nathan Estifanos, Adamu Addissie, Rahel Argaw Kebede, Amsalu Bekele.

**Investigation:** Nathan Estifanos.

**Methodology:** Nathan Estifanos, Gudina Egata, Negussie Deyessa.

**Project administration:** Nathan Estifanos.

**Resources:** Nathan Estifanos, Adamu Addissie, Rahel Argaw Kebede, Amsalu Bekele.

**Software:** Nathan Estifanos.

**Supervision:** Nathan Estifanos, Gudina Egata, Adamu Addissie, Rahel Argaw Kebede, Amsalu Bekele, Negussie Deyessa.

**Validation:** Nathan Estifanos, Gudina Egata, Adamu Addissie, Rahel Argaw Kebede, Amsalu Bekele, Negussie Deyessa.

**Visualization:** Nathan Estifanos.

**Writing – original draft:** Nathan Estifanos.

**Writing – review & editing:** Nathan Estifanos, Gudina Egata, Adamu Addissie, Rahel Argaw Kebede, Amsalu Bekele, Negussie Deyessa.

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
