## [Decision Letter · Decision Letter 0]

30 Jun 2025

Dear Dr. Estifanos,

Thank you for submitting your manuscript to PLOS ONE. After careful consideration, we feel that it has merit but does not fully meet PLOS ONE’s publication criteria as it currently stands. Therefore, we invite you to submit a revised version of the manuscript that addresses the points raised during the review process.

We look forward to receiving your revised manuscript.

Kind regards,

Xiong Xingyu, M.D.

Academic Editor

PLOS ONE

Journal Requirements:

Reviewers' comments:

Reviewer's Responses to Questions

**Comments to the Author**

1. Is the manuscript technically sound, and do the data support the conclusions?

Reviewer #1: Yes

Reviewer #2: Yes

Reviewer #3: Yes

2. Has the statistical analysis been performed appropriately and rigorously?

Reviewer #1: Yes

Reviewer #2: Yes

Reviewer #3: No

3. Have the authors made all data underlying the findings in their manuscript fully available?

Reviewer #1: No

Reviewer #2: Yes

Reviewer #3: Yes

4. Is the manuscript presented in an intelligible fashion and written in standard English?

Reviewer #1: Yes

Reviewer #2: Yes

Reviewer #3: No

Reviewer #1: The authors did a great job assembling a large, population-based dataset and applying a rigorous approach to a public health gap in lung cancer symptom awareness in Ethiopia.

Here are my remarks:

Abstract

1) Lines 32-33: For the aim, consider rewriting to a single clear objective sentence (such as "to identify latent classes of symptom awareness and their predictors”).

2) Line 50: Conclusion states “one-fourth (25 %)”; “one-fourth” already implies 25 %. Delete duplication.

Introduction

3) Line 65: Add citation for WHO guideline.

4) Lines 78–111: These three background paragraphs are dense and contain several ideas that are later repeated. Please condense them into 3–4 focused sentences that (i) summarise the burden of lung-cancer symptoms in Ethiopia, (ii) note the need of local awareness data, and (iii) introduce latent-class analysis as a novel approach. Merge this concise version with the end of the third paragraph (line 77).

Methods

5) Line 139: Include a citation for the formula.

6) Line 194-196: Collapsing current & former smokers into “ever” obscures an important gradient; mention rationale.

7) Line 253: Add a statement on how missing data was handled.

Results

8) Lines 317–318: Add a phrase on non-responders (n = 47).

Discussion

9) Line 546-548: Add citation to support speculation related to difference in findings for males vs. females.

10) Discussion is very lengthy, condense repetitive comparisons with prior studies.

Conclusion:

11) Delete details on subgroups for each category (such as "(secondary, diploma and vocational, and degree and above)") and condense findings to two sentences before policy implications.

General comments:

12) Ensure consistent decimal places for p-values throughout manuscript (preferably 2 decimal places except <0.001).

Reviewer #2: To identify latent classes within the population based on their levels of lung cancer symptoms awareness and predictors of class membership, in this manuscript, the authors conducted a face-to-face interview among 2388 adults in Addis Ababa, Ethiopia, for Lung Cancer Awareness Measurement. They found that only one fourth (25%) of the study participants had good awareness based on lung cancer awareness (LCA), and that respiratory lung cancer symptoms are more known than non-respiratory. The authors suggest that there is a need for class-specific educational intervention to raise awareness and focus on the less-known symptoms to reduce the likelihood of lung cancer being detected too late. These findings are interesting and significant. The manuscript is well written and the research data is presented in a logical way. Before publication of this manuscript in PLoS One, the following issues should be addressed:

Specific issues:

1. Authors should analyze and compare the differences of the lung cancer symptoms awareness between males and females in more detail with statistics and provide explanations.

2. Authors should also compare the differences of lung cancer symptoms awareness between smokers and non-smokers in more detail with statistics.

Reviewer #3: This research provides new insights into the knowledge of lung cancer symptoms in Ethiopia.

It aimed to identify population subgroups according to their level of lung cancer symptom knowledge as well as to obtain predictors of class membership using LCA among adults in Addis Ababa, the capital of Ethiopia.

The great utility of their results lies in the fact that such inferences can provide baseline evidence for the design of future intervention strategies.

However, the presentation of the manuscript and the development of the sections should be reviewed and improved before publication.

Main issues reviewed:

1) Abstract: In this section, lines 42 to 49 include only one sentence! Please shorten the sentences and write them correctly.

2) Introduction: It seems to be a good section for this topic, emphasizing various aspects of lung cancer, the risk factors associated with its incidence, some gaps in the literature about approaches to measurements of prior knowledge of early-stage disease,....etc. However, the authors should try to reduce its length in, for example, 4 important paragraphs.

In the abstract, lines 42 to 49 include only one sentence! Please shorten the sentences and write correctly.

It is too long, unnecessarily repetitive.

3) Methods: i) Air pollution, particularly particulate matter (PM2.5), is recognized as one significant risk factor for lung cancer in Ethiopia, contributing to increased hospitalizations and premature deaths. Both outdoor and indoor air pollution exacerbate this risk. In the research by Demeke Endalie et al (PLOS Digit Health. 2023 doi: 10.1371/journal.pdig.0000308,

"Analysis of lung cancer risk factors from medical records in Ethiopia using machine learning") were identified two important factors for lung cancer: air pollution and obesity, with significance weights of 0.21 and 0.14, respectively.

Considering the aforementioned is that my concern about the level of inference (and generality) of this work to the entire country is the level of generality is incorrect and exaggerated. Also, according to official data from 2023, the Ethiopian rural population accounts for about 78% of the total population. This means that the majority of Ethiopians live in rural areas, with agricultural production. Several papers have reported higher lung cancer incidence (and mortality) rates in rural areas compared to urban areas. Furthermore, specific research suggests that this disparity is related to factors such as higher prevalence of smoking, obesity, poor living conditions, limited access to health care and diagnostic centers, among others.

So:

-What can the authors say about the inferences made in this manuscript for the whole country?

-How might those factors or conditions (cited above) confound the context of residents' prior knowledge about lung cancer? This would basically affect the levels of lung cancer symptoms awareness.

The authors should consider this aspect and justify the generality of the work.

ii) line 133. The authors should rewrite this sentence describing the sample correctly.

iii) line 141: Proportion is a number P, 0=<p<=1! correct=" " please=" " this.=" ">iv) line 140-141: "Since the proportion of lung cancer awareness in our context was unknown, a conservative assumption of P = 50% was made". Please explain this choice better. In other countries, there are some papers with very different proportions that can serve as a reference. In my opinion, being quite conservative leads to smaller sample sizes. Please justify.

3) Results:

i) Figures 1 - 2 require better definition.

ii) The authors should emphasize the essence of the results presented in the tables and not describe them by reproducing them. Please rewrite some paragraphs since there is no need for so much explanation!

5) Discussion: i) It is too extensive, addressing too many aspects (and others) of the work. In doing so, it suffers from too many generalizations. Also, it reproduces numerical results already presented in the previous section (what for? they have already been read...).

The authors should focus on the main results and analyze why they contribute to creating new knowledge, i.e., identify distinct classes of population based on their level of knowledge of lung cancer symptoms and predictors of class membership. Rewrite please.

ii) line 631.633: "However, the study has its limitations. Given the cross-sectional design of the survey,

it is difficult to establish causality.

It is not difficult, it does not exist as such! Causality studies have a different methodological design.

iii) line 632-633: "In addition, the outcome of interest was measured through using spontaneous response questions, which may lead to overestimates by offering the possibility of guessing"

Just those limitations? the authors should comment on issues related to the type(s) of questionnaire used, the nature of the response (self-reported), etc, etc.

6) Conclusion: It is a succinct restatement of the results. There should be a synthesis of the results, and this should be presented in its entirety and in the context of the study. Rewrite.</p<=1!>

**Do you want your identity to be public for this peer review?** For information about this choice, including consent withdrawal, please see our Privacy Policy

Reviewer #1: No

Reviewer #2: No

Reviewer #3: No

---

## [Author Response · Author response to Decision Letter 1]

31 Jul 2025

Response to the Editor’s and Reviewers’ Comments

Dear Editor,

Thank you for your valuable feedback and the opportunity to revise our manuscript. Your guidance, along with the detailed and constructive comments from the reviewers, has been instrumental in improving the clarity, quality, and scientific value of our work. We have carefully addressed both the editor’s and reviewers’ comments. Additionally, we have extensively edited the manuscript. Below is a point-by-point response to the editor’s and reviewers’ comments.

Journal Requirements:

Editor Comment # 1

Response: We have adhered to PLOS ONE’s formatting guidelines. The main text follows the structure and formatting shown in the official sample templates provided by the journal.

Comment # 2

Response: Thank you for pointing this out. We have corrected and aligned the grant numbers and funding details in both the ‘Funding Information’ and ‘Financial Disclosure’ sections to ensure consistency.

Comment # 3

Response: Thank you for your critical concern. We have updated the Data Availability Statement as “All data are available within the manuscript and its supporting materials”.

Reviewer comments

Reviewer #1

Comment #1: The authors did a great job assembling a large, population-based dataset and applying a rigorous approach to a public health gap in lung cancer symptom awareness in Ethiopia. Here are my remarks:

Response: We sincerely thank the reviewer for their positive assessment of our work. We appreciate the recognition of the scope and rigor of our study and have carefully considered all subsequent remarks to further improve the manuscript.

Abstract

Comment #2: 1) Lines 32-33: For the aim, consider rewriting to a single clear objective sentence (such as "to identify latent classes of symptom awareness and their predictors”).

Response: Thank you for your suggestion. We have revised the objective into a single, clear sentence (lines 31–32).

Comment #3: 2) Line 50: Conclusion states “one-fourth (25 %)”; “one-fourth” already implies 25 %. Delete duplication.

Response: Thank you for the suggestion. As recommended, we have removed the redundant percentage on line 45.

Introduction

Comment #4: 3) Line 65: Add citation for WHO guideline.

Response: Thank you for this valuable suggestion. We have added the citation for the WHO guideline, which is now listed as reference [4].

Comment #5: 4) Lines 78–111: These three background paragraphs are dense and contain several ideas that are later repeated. Please condense them into 3–4 focused sentences that (i) summarise the burden of lung-cancer symptoms in Ethiopia, (ii) note the need of local awareness data, and (iii) introduce latent-class analysis as a novel approach. Merge this concise version with the end of the third paragraph (line 77).

Response: Thank you for your constructive comment. In response, we have revised the paragraphs accordingly; please refer to lines 64–83.

Methods

Comment #6: 5) Line 139: Include a citation for the formula.

Response: Thank you. We have added the citation for Cochran’s single population proportion formula, which is now included as reference [37] on line 109.

Comment #7: 6) Line 194-196: Collapsing current & former smokers into “ever” obscures an important gradient; mention rationale.

Response: Thank you for the insightful comment. We agree that current and former smokers may have different levels of awareness. However, due to the very low prevalence of smoking in our study population—particularly among females—further disaggregation would have resulted in small cell sizes and unstable estimates. Therefore, we combined current and former smokers into a single “ever smoker” category to preserve statistical power. We have now clarified this rationale in the revised manuscript. Please refer to lines 168–170.

Comment #8: 7) Line 253: Add a statement on how missing data was handled.

Response: Thank you. We have added a statement to clarify that no missing data were encountered, as all responses were complete during the face-to-face interview process. Please refer to line 214.

Results

Comment #9: 8) Lines 317–318: Add a phrase on non-responders (n = 47).

Response: Thank you for your insightful comment. We have added a statement clarifying that the 47 individuals were those who were unavailable at home despite three visit attempts. Please refer to line 294. They were not replaced to preserve the integrity of the probability sampling design and to avoid potential selection bias.

Discussion

Comment #10: 9) Line 546-548: Add citation to support speculation related to difference in findings for males vs. females.

Response: Thank you for the valuable comment. We have added relevant citations to support our discussion on sex differences. The citation is now included as reference [52] on line 459.

Comment #11: 10) Discussion is very lengthy, condense repetitive comparisons with prior studies.

Response: Thank you for your constructive feedback. We have carefully revised the Discussion section as recommended, reducing it by two and a half pages. Please refer to pages 18–21.

Conclusion:

Comment #12: 11) Delete details on subgroups for each category (such as "(secondary, diploma and vocational, and degree and above)") and condense findings to two sentences before policy implications.

Response: Thank you for your helpful comment. We have revised the conclusion based on your recommendation.

General comments:

Comment #13: 12) Ensure consistent decimal places for p-values throughout manuscript (preferably 2 decimal places except <0.001).

Response: Thank you for this helpful suggestion. We have reviewed the manuscript and standardized the formatting of all p-values as recommended. Specifically, we applied the following rules for consistent reporting: p > 0.01 is reported to two decimal places; 0.01 ≥ p > 0.001 is reported to three decimal places; and p ≤ 0.001 is reported as p < 0.001.

Reviewer #2

Comment 1: Reviewer #2:

To identify latent classes within the population based on their levels of lung cancer symptoms awareness and predictors of class membership, in this manuscript, the authors conducted a face-to-face interview among 2388 adults in Addis Ababa, Ethiopia, for Lung Cancer Awareness Measurement. They found that only one fourth (25%) of the study participants had good awareness based on lung cancer awareness (LCA), and that respiratory lung cancer symptoms are more known than non-respiratory. The authors suggest that there is a need for class-specific educational intervention to raise awareness and focus on the less-known symptoms to reduce the likelihood of lung cancer being detected too late. These findings are interesting and significant. The manuscript is well written and the research data is presented in a logical way. Before publication of this manuscript in PLoS One, the following issues should be addressed:

Response: We sincerely thank the reviewer for the thoughtful and encouraging comments. We appreciate the positive feedback regarding the study’s design, data presentation, and the significance of our findings. We have carefully considered the specific issues raised and made the necessary revisions to improve the manuscript accordingly.

Specific issues:

Comment #2: 1. Authors should analyze and compare the differences of the lung cancer symptoms awareness between males and females in more detail with statistics and provide explanations.

Response: We thank the reviewer for this valuable insight. As outlined in the manuscript, we analyzed and compared lung cancer symptom awareness between males and females by including sex as a predictor in the latent class multinomial logistic regression model. Our findings indicate that males had significantly higher awareness of lung cancer symptoms compared to females, reflected by their higher likelihood of belonging to the “good awareness” class. Specifically, males were more likely than females to be in the “good awareness” class compared to the “low awareness” class (AOR = 1.66; 95% CI: 1.13–2.43; p = 0.001). This analysis is presented in the Results section (Page 15, Lines 381–383) and Table 4, and is further discussed in the Discussion section (Page 19, Lines 453–460).

Comment #3: 2. Authors should also compare the differences of lung cancer symptoms awareness between smokers and non-smokers in more detail with statistics.

Response: We thank the reviewer for this thoughtful comment. We included smoking status as a predictor in the latent class multinomial logistic regression model to assess differences in lung cancer symptom awareness between smokers and non-smokers. The findings revealed no statistically significant difference in awareness between the two groups. Although smokers were slightly more likely than non-smokers to belong to the “good awareness” class compared to the “low awareness” class, this association was not statistically significant (AOR = 1.14; 95% CI: 0.59–2.17; p = 0.70). These findings are reported in the Results section (Page 15, Lines 404–406) and Table 4, and briefly discussed in the Discussion section (Page 20, Lines 493–498).

Reviewer #3

Comment #1:Reviewer #3: This research provides new insights into the knowledge of lung cancer symptoms in Ethiopia.

It aimed to identify population subgroups according to their level of lung cancer symptom knowledge as well as to obtain predictors of class membership using LCA among adults in Addis Ababa, the capital of Ethiopia.

The great utility of their results lies in the fact that such inferences can provide baseline evidence for the design of future intervention strategies.

However, the presentation of the manuscript and the development of the sections should be reviewed and improved before publication.

Response: We thank the reviewer for their thoughtful and encouraging feedback. We are pleased that the research and its findings were considered valuable for informing future intervention strategies. In response to the comments, we have thoroughly revised the manuscript to improve clarity, structure, and presentation. Revisions include condensing repetitive content and improving the flow of the Introduction, Methods, Results, and Discussion sections. Below, we provide a point-by-point response to each comment.

Main issues reviewed:

Comment #2: 1) Abstract: In this section, lines 42 to 49 include only one sentence! Please shorten the sentences and write them correctly.

Response: Thank you for pointing this out. We have revised the sentence to enhance readability and clarity by restructuring it into a clear and concise format. Please refer to lines 41–44.

Comment #3: 2) Introduction: It seems to be a good section for this topic, emphasizing various aspects of lung cancer, the risk factors associated with its incidence, some gaps in the literature about approaches to measurements of prior knowledge of early-stage disease,....etc. However, the authors should try to reduce its length in, for example, 4 important paragraphs.

In the abstract, lines 42 to 49 include only one sentence! Please shorten the sentences and write correctly.

It is too long, unnecessarily repetitive.

Response: Thank you for your valuable input. We have revised the Introduction section as you recommended. Please refer to the updated Introduction section.

Comment #4: 3) Methods: i) Air pollution, particularly particulate matter (PM2.5), is recognized as one significant risk factor for lung cancer in Ethiopia, contributing to increased hospitalizations and premature deaths. Both outdoor and indoor air pollution exacerbate this risk. In the research by Demeke Endalie et al (PLOS Digit Health. 2023 doi: 10.1371/journal.pdig.0000308,

"Analysis of lung cancer risk factors from medical records in Ethiopia using machine learning") were identified two important factors for lung cancer: air pollution and obesity, with significance weights of 0.21 and 0.14, respectively.

Considering the aforementioned is that my concern about the level of inference (and generality) of this work to the entire country is the level of generality is incorrect and exaggerated. Also, according to official data from 2023, the Ethiopian rural population accounts for about 78% of the total population. This means that the majority of Ethiopians live in rural areas, with agricultural production. Several papers have reported higher lung cancer incidence (and mortality) rates in rural areas compared to urban areas. Furthermore, specific research suggests that this disparity is related to factors such as higher prevalence of smoking, obesity, poor living conditions, limited access to health care and diagnostic centers, among others.

So:

-What can the authors say about the inferences made in this manuscript for the whole country?

-How might those factors or conditions (cited above) confound the context of residents' prior knowledge about lung cancer? This would basically affect the levels of lung cancer symptoms awareness.

The authors should consider this aspect and justify the generality of the work.

Comment #5: Thank you for this insightful observation. We acknowledge that our study was conducted in Addis Ababa and, therefore, primarily reflects the awareness level of an urban population. We do not intend to generalize the findings to the entire country, especially rural areas where access to health information, services, and exposure to risk factors may differ significantly, as you clearly described. However, we believe that the awareness level observed in Addis Ababa can still offer valuable insights. As the capital city, with relatively better access to health services, information, and infrastructure, Addis Ababa is expected to have comparatively higher lung cancer awareness than rural areas. Therefore, the low awareness observed in this urban setting likely reflects an even more concerning situation in less-served regions. We have included this limitation in the Discussion section (Page 20, lines 507–510) and recommend further research in rural populations to fully understand the national picture (Page 21, lines 521–522).

Comment #6: ii) line 133. The authors should rewrite this sentence describing the sample correctly.

Response: Thank you for your insightful comment. We have revised the manuscript accordingly (lines 101–104).

Comment #7: iii) line 141: Proportion is a number P, 0=iv) line 140-141: "Since the proportion of lung cancer awareness in our context was unknown, a conservative assumption of P = 50% was made". Please explain this choice better. In other countries, there are some papers with very different proportions that can serve as a reference. In my opinion, being quite conservati

---

## [Decision Letter · Decision Letter 1]

8 Sep 2025

Lung cancer symptoms awareness among Ethiopian adults, Ethiopia: Latent class analysis

PONE-D-25-05177R1

Dear Dr. Estifanos,

We’re pleased to inform you that your manuscript has been judged scientifically suitable for publication and will be formally accepted for publication once it meets all outstanding technical requirements.

Kind regards,

Xingyu Xiong, Ph.D.

Academic Editor

PLOS ONE

Additional Editor Comments (optional):

Reviewers' comments:

Reviewer's Responses to Questions

**Comments to the Author**

Reviewer #2: All comments have been addressed

Reviewer #3: (No Response)

2. Is the manuscript technically sound, and do the data support the conclusions?

Reviewer #2: Yes

Reviewer #3: Yes

3. Has the statistical analysis been performed appropriately and rigorously?

Reviewer #2: Yes

Reviewer #3: Yes

4. Have the authors made all data underlying the findings in their manuscript fully available?

Reviewer #2: Yes

Reviewer #3: Yes

5. Is the manuscript presented in an intelligible fashion and written in standard English?

Reviewer #2: Yes

Reviewer #3: Yes

Reviewer #2: Authors addressed my comments properly with results and discussion in revised manuscript. I highly appreciate authors' good response.

Reviewer #3: The authors have submitted a thorough revision of the manuscript, including corrections to the discussion (two important items) and conclusion. These two aspects were central to the original recommendation.

**Do you want your identity to be public for this peer review?** For information about this choice, including consent withdrawal, please see our Privacy Policy

Reviewer #2: No

Reviewer #3: **Yes: ** Maria del Pilar Diaz

---

## [Editor Report · Acceptance letter]

PONE-D-25-05177R1

PLOS ONE

Dear Dr. Estifanos,

I'm pleased to inform you that your manuscript has been deemed suitable for publication in PLOS ONE. Congratulations! Your manuscript is now being handed over to our production team.

Kind regards,

on behalf of

Dr. Xingyu Xiong

Academic Editor

PLOS ONE